# Protective Effects of *Centella asiatica* on Cognitive Deficits Induced by D-gal/AlCl_3_ via Inhibition of Oxidative Stress and Attenuation of Acetylcholinesterase Level

**DOI:** 10.3390/toxics7020019

**Published:** 2019-03-30

**Authors:** Samaila Musa Chiroma, Mohamad Taufik Hidayat Baharuldin, Che Norma Mat Taib, Zulkhairi Amom, Saravanan Jagadeesan, Mohd Ilham Adenan, Onesimus Mahdi, Mohamad Aris Mohd Moklas

**Affiliations:** 1Department of Human Anatomy, Faculty of Medicine and Health Sciences, Universiti Putra Malaysia, 43400 Serdang, Selangor, Malaysia; musasamailachiroma@yahoo.com (S.M.C.); taufikb@upm.edu.my (M.T.H.B.); chenorma@upm.edu.my (C.N.M.T.); mljsaravanan@gmail.com (S.J.); omahdi2010@gmail.com (O.M.); 2Department of Human Anatomy, Faculty of Basic Medical Sciences, University of Maiduguri, 600230 Maiduguri, Borno State, Nigeria; 3Faculty of Health Sciences, Universiti Teknologi Mara (UiTM) Kampus Puncak Alam, 42300 Bandar Puncak Alam, Selangor, Malaysia; zulkha2992@puncakalam.uitm.edu.my; 4Department of Human Anatomy, Universiti Tunku Abdul Rahman (UTAR), Bandar Sungai Long, 43000 Kajang, Selangor, Malaysia; 5Atta-ur-Rahman Institute for Natural Product Discovery, Universiti Teknologi Mara (UiTM) Kampus Puncak Alam, 42300 Bandar Puncak Alam, Selangor, Malaysia; mohdilham@puncakalam.uitm.edu.my; 6Department of Human Anatomy, College of Medical Sciences, Gombe State University, 760211 Gombe, Gombe State, Nigeria

**Keywords:** *Centella asiatica*, cognitive deficit, Alzheimer’s disease, morphological aberration, acetylcholinesterase, oxidative stress

## Abstract

Alzheimer’s disease (AD) is a progressive neurodegenerative disorder with cholinergic dysfunctions and impaired redox homeostasis. The plant *Centella asiatica* (CA) is renowned for its nutritional benefits and herbal formulas for promoting health, enhancing cognition, and its neuroprotective effects. The present study aims to investigate the protective role of CA on D-gal/AlCl_3_-induced cognitive deficits in rats. The rats were divided into six groups and administered with donepezil 1 mg/kg/day, CA (200, 400, and 800 mg/kg/day) and D-gal 60 mg/kg/day + AlCl_3_ 200 mg/kg/day for 10 weeks. The ethology of the rats was evaluated by the Morris water maze test. The levels of acetylcholinesterase (AChE), phosphorylated tau (P-tau), malondialdehyde (MDA) and activities of superoxide dismutase (SOD), in the hippocampus and cerebral cortex were estimated by enzyme-linked immunosorbent assay (ELISA). Additionally, the ultrastructure of the prefrontal cortex of the rats’ was observed using transmission electron microscopy (TEM). Rats administered with D-gal/AlCl_3_ exhibited cognitive deficits, decreased activities of SOD, and marked increase in AChE and MDA levels. Further, prominent alterations in the ultrastructure of the prefrontal cortex were observed. Conversely, co-administration of CA with D-gal/AlCl_3_ improved cognitive impairment, decreased AChE levels, attenuated the oxidative stress in hippocampus and cerebral cortex, and prevented ultrastructural alteration of neurons in the prefrontal cortex. Irrespective of the dose of CA administered, the protective effects were comparable to donepezil. In conclusion, this study suggests that CA attenuated the cognitive deficits in rats by restoring cholinergic function, attenuating oxidative stress, and preventing the morphological aberrations.

## 1. Introduction

Alzheimer’s disease (AD) is a progressive neurodegenerative disorder associated with cholinergic dysfunction and impaired redox homeostasis in the brain [1]. The etiology of AD is associated with the deficiency of a neurotransmitter acetylcholine (ACh), which is involved in the communication between neurons in the brain [2]. Acetylcholinesterase (AChE) is an enzyme in the brain which breaks down ACh into inactive metabolites choline and acetate, and thus the increased activity of this enzyme can result in a deficiency of ACh. Drugs (AChE inhibitors) that were able to restore the appropriate levels of ACh were developed on the rationale of cholinergic hypothesis of AD, wherein these drugs act by inhibiting the action of AChE [3]. Therefore, AChE inhibitors are still the mainstay of AD treatment though they do not cure the disease, but play a significant role in palliative management of AD [4]. Another pathological hallmark of AD is the neurofibrillary tangles in the brain which are formed by hyperphosphorylation and abnormal aggregation of tau protein [5]. It has been hypothesized that the aggregates of tau in the brains of patients with AD disrupts the microtubule formation and maintenance, damaging axonal transport in the neuronal cells, which results in synaptic starvation and neuronal death [6,7,8]. Accumulating evidence from various studies on tau phosphorylation and neurofibrillary tangles have kindled the need to focus attention on tau as a therapeutic target in the treatment of AD and other tauopathies. The current drugs that are based on amyloid beta (Aβ) have not yielded the desired results, prompting clinicians and researchers to look for alternatives drugs, including phosphorylated tau (P-tau) based therapeutics [5,9].

D-galactose (D-gal) is present in the human body as a reducing sugar, but when it exceeds its normal level, it is oxidized into hydrogen peroxide (H_2_O_2_) and aldehydes by galactose oxidase [10]. Animals, when treated chronically with D-gal, exhibit aging-related changes such as decreased activities of antioxidant enzymes, increased levels of oxidants, and cognitive impairments [11,12]. Further, intraperitoneal injection (i.p) of D-gal affects the cholinergic system resulting in increased levels of AChE in the brains of rats [13]. Aluminum (Al), is a naturally occurring toxic trace element on earth [12], which has been linked to pathogenesis of AD [14]. Kumar [15] reported an upsurge in AChE activities, oxidative damage, and cognitive dysfunction in rats after chronic administration of aluminum chloride (AlCl_3_). Accumulating evidence showed that co-administration of D-gal and AlCl_3_ to rats impaired their cognitive functions, increased AChE activities, altered oxidative balance, and induced neurodegeneration [16,17,18]. Therefore, rats chronically administered with D-gal and AlCl_3_ could be a good model for studying AD-related pathologies and for screening of anti-AD drugs.

Oxidative stress being an essential component in the pathogenesis of AD, designing multifunctional agents including antioxidative properties for targeting AD is crucial. Several researchers have tried to develop and test dual functioning drugs, which comprises of both “oxidative stress suppressing” moiety and an “acetylcholinesterase inhibitory” moiety [19,20]. The plant *Centella asiatica* (CA) also known as “Icudwane” in South Africa, “Gotu Kola” in India or “Indian pennywort” in USA has several medicinal properties which includes improvement of cognition and wound healing abilities [21]. The neuroprotective effects of CA which are seen in animal disease models could be attributed to its antioxidant properties [22]. CA has also shown to have numerous other pharmacological properties like analgesic and anti-inflammatory properties [23] and anti-hyperglycemic properties on obese diabetic rats [24]. Kumar and Gupta reported that the aqueous extract of CA has two prominent effect on the brain that is improving learning and memory and antioxidant properties [25].

Data from previous research indicated that CA attenuated cognitive impairments in D-gal- and AlCl_3_-induced rats through the prevention of hippocampal neuronal death and maintenance of its ultrastructure [26]. Whether CA can also reduce AChE levels and prevent oxidative stress to attenuate cognitive decline in D-gal- and AlCl_3_-induced rats remains unknown. Hence, the current work aimed to study the protective properties of CA on cognition by subjecting rats to the Morris water maze (MWM) test. Subsequent to the behavioral tests, the hippocampal and cerebral cortex tissues of the rats were analyzed for AChE, P-tau, and malondialdehyde (MDA) levels, as well as superoxide dismutase (SOD) activities, besides the evaluation of the ultrastructure of their prefrontal cortex using transmission electron microscopy (TEM).

## 2. Materials and Methods

### 2.1. Materials

The chemical donepezil hydrochloride was purchased from Esai Co. Ltd. (Tokyo, Japan), D-gal and AlCl_3_ were bought from Sigma Aldrich (St. Louis, MO, USA), while CA extract (Reference number: AuRins-MIA-1-0) [27,28] was obtained from Atta-ur-Rahman Institute for Natural Product Discovery, Universiti Teknology Mara (UiTM) Puncak Alam, Selangor, Malaysia. Enzyme-linked immunosorbent assay (ELISA) kits for AChE, SOD, and P-Tau were purchased from Elabscience (Houston, TX, USA), while MDA kit was purchased from Cayman Chemical Company (Ann Arbor, MI, USA). The tank used for the MWM and the video camera (Logitech) were obtained locally in Malaysia while ANY-maze software (Version 5.32, Stoelting Co., Wood Dale, IL, USA, 2018) was used for the behavioral analysis. A transmission electron microscope (TEM) TEMLEO LIBRA-120 was used to view the samples.

### 2.2. Animals

Thirty-six male albino Wistar rats that were 2–3 months old and weighing 200–250 g, were used in the experiment. The study and protocol followed was approved by Institutional Animal Care and Use Committee, Universiti Putra Malaysia on 20th March 2017, with project identification code UPM/IACUC/AUP-R096/2016. The rats were housed 2–3 per cage in a temperature-controlled room at 22 ± 3 °C with 12 h light/dark cycles. The rats had free access to standard laboratory rat chow and water, and all the chemicals, drugs, and CA extract were administered in the morning and the behavioral tests were conducted in the afternoon.

### 2.3. Study Design

The rats were acclimatised for one week and then randomly divided into six groups comprising of six rats each. The doses of D-gal, AlCl_3_, donepezil and CA (Table 1) were selected based on previous works [18,29] and on published literatures researched [30,31]. Starting from the 10th week of administration, the rats were assessed in the MWM test while administration of drugs and CA extract was still going on, after which the rats were euthanized by decapitation in order to prevent their brain tissue from contamination by chemicals used, such as anesthetics and gases [32].

### 2.4. Morris Water Maze (MWM) Test

The MWM test described by Morris [33] has been accepted as a standard test for studying cognitive behaviors in rodents. The test is based on the rodents’ inherent tendency to avert swimming in water, and it requires the animals to use visual cues to learn to identify the position of a platform submerged in water which provides a route to escape [34]. The description of the basic apparatus, the procedure for the test, and tips of troubleshooting are available in previously published articles [18,33,34].

### 2.5. Sample Collection and Preparation

Rats were humanely euthanised at the end of the 11th week through decapitation, and the brain samples were rapidly harvested, rinsed in cold saline to wash excess blood, and the brain samples for histopathological study were fixed in 10% formalin while the remaining brain samples were stored at −80 °C until assayed.

### 2.6. Preparation of Brain Homogenate

To prepare brain homogenate, the hippocampus and cerebral cortex from the brain samples kept at −80 °C were taken and rapidly sliced into small pieces on a cold plate. For ELISA, the tissues were weighed and homogenized in cold phosphate buffered saline (PBS) (tissue weight (g): PBS (mL) volume 1:9) using a tissue homogenizer. The homogenates were then centrifuged at 5000 or 10,000× *g* for 5 min to get the supernatant.

### 2.7. Protein Estimation

The concentration of protein in the hippocampus and cerebral cortex were measured using the bicinchoninic assay (BCA assay). The standard used was bovine serum albumin (BSA) (2 mg/mL) with a working range of 20–2000 µg/mL.

### 2.8. Enzyme Linked Immunosorbent Assay (ELISA)

#### 2.8.1. AChE, P-tau, and SOD

The levels of AChE and P-tau and the activities of SOD from the hippocampal and cerebral cortex homogenates were analyzed using quantitative sandwich ELISA technique, with strict adherence to the manufacturer’s manual from Elabscience Biotechnology Inc. (Houston, TX, USA).

#### 2.8.2. MDA

The levels of MDA were measured according to the manufacturer’s guide from Cayman Chemical Company, (Ann Arbor, MI, USA). MDA level was assayed by monitoring thiobarbituric acid (TBA) reactive substance formation. A volume of 100 µL of each brain homogenate was mixed with 100 µL of sodium dodecyl sulphate (SDS) followed by the addition of 4 mL color reagent (530 mg of TBA, 50 mL of diluted TBA acetic acid solution, and 50 mL of diluted sodium hydroxide solution) were forced down the vile. The mixture was then kept in a vigorously boiling water bath for 1 h and then cooled immediately on ice for 10 min. The vile was then centrifuged at 1600× *g* for 10 min at 4 °C, after which 150 µL of the supernatant were aliquoted in duplicate into a 96 well microplate and read at 532 nm using a spectrophotometer. The MDA levels were expressed as nmol per mg of protein.

### 2.9. Transmission Electron Microscopy (TEM)

#### 2.9.1. Tissue Preparation and Qualitative Analysis

TEM was used to assess the ultrastructure of the prefrontal cortex of the rat’s brain; the method followed has been previously described [26].

#### 2.9.2. Mitochondrial Abnormalities

Based on the TEM appearance the neuronal mitochondria were categorised into damaged (disrupted outer membrane and fragmented cristae) or healthy (continuous outer membrane and compacted matrix) [35]. The mitochondrial abnormalities were further categorised as: 1. Swollen dense mitochondria; 2. Swollen clear mitochondria; 3. Dark degenerated mitochondria; and 4. Fragmentation of cristae or disjointed outer or inner mitochondrial membrane [36].

#### 2.9.3. Abnormalities of the Nucleus

Several abnormalities were identified in the nucleus, including pyknosis, degeneration of chromatin material, disruption of the nuclear membrane, degeneration of the nucleoli, and crescent formation.

#### 2.9.4. Synaptic Abnormalities

It has been previously observed that the ultrastructural abnormalities of synapses included irregular synaptic connections, widened synaptic cleft, reduced synaptic vesicles, and decreases in the number of synapses [26]. The other abnormalities observed are disruption of pre- and post-synaptic mitochondria, rupture of synaptic vesicles, and thinning of post synaptic density.

### 2.10. Statistical Analysis

The results of the place navigation test aspect of the MWM test was analyzed using a two-way ANOVA, while that of the spatial probe test aspect of the MWM test was analyzed using a one-way ANOVA. The remaining data were analyzed by one-way ANOVA, while Tukey’s post hoc comparison was used where applicable. *p* < 0.05 were considered significant and results were presented as mean ± SEM. GraphPad Prism version 6 (ISI, San Diego, CA, USA) software was used for the for the data analysis.

## 3. Results

### 3.1. Protective Effects of CA on Cognitive Functions

The MWM test was carried out to evaluate the effects of CA on cognitive functions. Figure 1A,B shows the average escape latency to locate the submerged escape platform for each day of the test and the average distance covered by the rat groups for the five-day test, respectively. The two-way ANOVA showed a statistically significant interaction between the effect of treatment and days of treatment, [F(5, 30) = 10.45, *p* = 0.0001] in the escape latency of the rats. Tukey’s post hoc comparison showed a statistically significant decrease (*p* < 0.05) in time to locate the submerged escape platform on day four and day five by the control, donepezil, CA 200, CA 400, and CA 800 groups of rats when compared to the model group. Further, the two-way ANOVA showed a significant interaction between the effect of treatment and the days of treatment, [F(5, 30) 7.180, *p* = 0.0002] on the average distance covered by the rat groups in search of the escape platform. Tukey’s post hoc showed a statistically significant increase (*p* < 0.05) in the distance covered by the model-group rats on day four as compared to the rats in the control, donepezil, CA 200, and CA 800 groups. Subsequently, a similar trend was also observed on day five of the test where a statistically significant increase was observed in the distance covered by rats of the model group when compared to the control, donepezil, and CA (200, 400, and 800) groups of rats. Figure 1C shows the probe test, in which the one-way ANOVA [F(5, 30) = 13.21, *p* = 0.0001] showed significant differences among the rat groups. Tukey’s post hoc comparison showed a statistically significant increased time spent in the target quadrant (quadrant of the tank where the submerged escape platform was present) by the control (16 ± 3.95, *p* = 0.0001), donepezil (14.0 ± 3.10, *p* = 0.0001), CA 200 (12.1 ± 2.32, *p* = 0.001), CA 400 (15.5 ± 1.2, *p* = 0.0001), and CA 800 (14.9 ± 3.63, *p* = 0.0001) groups of rats compared to the model group of rats (5.1 ± 1.27).

### 3.2. Effects of CA on AChE Level

The ELISA results (Figure 2A) of AChE levels in the hippocampus of the rat groups revealed a statistically significant differences as shown by the one-way ANOVA [F(5, 12) = 14.86, *p* = 0.0001]. Tukey’s post hoc showed a significant decrease in the levels of AChE in the control (0.6 ± 0.20, *p* = 0.0002), donepezil (0.5 ± 0.25, *p* = 0.0001), CA 200 (0.8 ± 0.16, *p* = 0.001), CA 400 (0.7 ± 0.12, *p* = 0.0005), and CA 800 (0.7 ± 0.20, *p* = 0.0003) groups of rats when compared to rats in the model group (1.9 ± 0.35).

A statistically significant differences were observed in the AChE levels in cerebral cortex (Figure 2B) among the groups of rats as shown by the one-way ANOVA [F(5, 12) = 22.02, *p* = 0.0001]. Tukey’s comparison indicated a decreased AChE levels in the control (1.8 ± 0.01, *p* = 0.0001), donepezil (1.9 ± 0.05, *p* = 0.0001), CA 200 (2.0 ± 0.19, *p* = 0.0003), CA 400 (1.5 ± 0.19, *p* = 0.0001), and CA 800 (1.8 ± 0.06, *p* = 0.0001) groups of rats when compared to rats in the model group (3.3 ± 0.28).

### 3.3. Effects of CA on P-Tau Level

To investigate the protective effects of CA on D-gal- and AlCl_3_-induced rats, the levels of P-tau in the rats’ brains were measured using ELISA. A one-way ANOVA [F(5, 12) = 11.37, *p* = 0.0003] showed statistically significant differences in P-tau levels in the hippocampus among the rat groups (Figure 3A). A significant rise in P-tau level was seen in the model group (34 ± 4.86, *p* = 0.0001) when compared to the control (21 ± 1.72), as shown by Tukey’s post hoc. Further, Tukey’s post hoc revealed a significant reduction of P-tau levels in the donepezil treated group (20 ± 2.39, *p* = 0.0007) when compared to the model group (34 ± 4.66). Although, no statistically significant differences were seen in the CA-treated groups. The one-way ANOVA [F(5, 12) = 18.09, *p* = 0.0001] showed statistically significant differences in P-tau levels in the cerebral cortex (Figure 3B) among the rat groups. Tukey’s post hoc comparison showed a significant rise in P-tau levels in the model group (32.6 ± 1.83, *p* = 0.0001) when compared to the rats in the control group (20 ± 2.20). Treating the rats with donepezil significantly reduced the levels of P-tau (22.5 ± 2.27, *p* = 0.0003) when compared to the model group (32.6 ± 1.83). Although, no statistically significant differences were seen in groups of rats treated with CA at different doses.

### 3.4. Effects of CA on MDA Level

As shown in Figure 4, the one-way ANOVA revealed statistically significant differences of MDA levels in the hippocampus and cerebral cortex among the rat groups [Hippocampus: F(5,12) = 17.23, *p* = 0.0002, cerebral cortex: F(5,12) = 20.28, *p* = 0.0001]. Tukey’s post hoc revealed significant increases of MDA levels in the hippocampus of rats induced with D-gal and AlCl_3_ (9.3 ± 1.68, *p* = 0.0002) when compared to the control group of rats (4.1 ± 0.90). Treatment with donepezil significantly reduced the MDA level (4.7 ± 0.57, *p* = 0.0005) in the hippocampus. Similarly, the MDA levels in D-gal- and AlCl_3_-induced rats were significantly reduced by administration of CA 200 (4.5 ± 0.64, *p* = 0.0004), CA 400 (3.7 ± 0.55, *p* = 0.0001), and CA 800 (3.4 ± 0.47, *p* = 0.0001) when compared to the model group (9.3 ± 1.68). No differences were observed between the control, donepezil, and CA-treated groups of rats. Consequently, Tukey’s post hoc comparison showed a significant increase of MDA levels (10.34 ± 0.96, *p* = 0.0001) in the cerebral cortex of D-gal- and AlCl_3_-induced rats when compared to the control (4.6 ± 0.57) group of rats. Further, treatment of D-gal- and AlCl_3_-induced rats with donepezil significantly reduced the MDA level in their cerebral cortex (3.8 ± 0.43, *p* = 0.0001). Similarly, marked reductions were also observed in MDA level in the cerebral cortex of D-gal- and AlCl_3_-induced rats co-administered with CA 200 (6.95 ± 1.50, *p* = 0.0006), CA 400 (5.5 ± 0.57, *p* = 0.0004, and CA 800 (4.7 ± 1.00, *p* = 0.0001) when compared to the model group (10.3 ± 0.96). To conclude, significant decreases in MDA levels were observed in the cerebral cortex of the donepezil-treated group (3.8 ± 0.43, *p* = 0.01) when compared with the CA 200-treated group of rats (6.95 ± 1.50).

### 3.5. The Effects of CA on SOD Activities

To assess the antioxidant effects of CA in D-gal- and AlCl_3_-induced rats, the activity of SOD in the rat’s hippocampus and cerebral cortex was measured. The results (Figure 5A) of SOD activity in the hippocampus of rats revealed statistically significant differences among the various rat groups, as shown by the one-way ANOVA [F(5, 12) = 7.345, *p* = 0.0023]. The Tukey’s post hoc showed statistically significant differences in the decrease in the activity of SOD in the model group (11.67 ± 1.5, *p* = 0.002) when compared to the control group of rats (21.33 ± 3.7). Whereas, increases in the activity of SOD was observed in the control (21.3 ± 3.7, *p* = 0.0002), donepezil (18.9 ± 1.60, *p* = 0.0197), CA 200 (18.6 ± 2.71, *p* = 0.025), CA 400 (20 ± 1.12, *p* = 0.0087), and CA 800 (21.1 ± 1.7, *p* = 0.0037) groups of rats, when compared to the model group of rats (11.6 ± 1.5).

Further, the one-way ANOVA showed statistically significant differences of SOD activities in the cerebral cortex [F(5, 12) = 14.14, *p* = 0.0001] (Figure 5B) among the various rat groups. Tukey’s post hoc showed statistically significant differences in the decrease of SOD activity in the model group (11.33 ± 1.5, *p* = 0.002) when compared to the control group of rats (21.0 ± 1). Meanwhile, increased SOD activity was observed in the control (23 ± 1.11, *p* = 0.001), donepezil (19 ± 1.02, *p* = 0.0026), CA 200 (18.3 ± 1.52, *p* = 0.00053), CA 400 (20.6 ± 2.5, *p* = 0.0005), and CA 800 (20 ± 2.64, *p* = 0.0009) groups of rats, when compared to the model group of rats (11.3 ± 1.52).

### 3.6. Ultrastructural Changes

TEM was used to access the effects of CA on the ultrastructure of the prefrontal cortex in D-gal- and AlCl_3_-induced rat’s models of AD.

#### 3.6.1. Mitochondria

The pictures in Figure 6 are electron microscopic images of the prefrontal cortex of rats from all groups. It can be observed that the rats of the control group showed healthy neuronal mitochondria which appear as ovoid or round shaped, with dense matrix, intact double mitochondrial membrane, and parallel-organized cristae (Figure 6A). Prominent mitochondrial aberrations of various degrees were seen in D-gal- and AlCl_3_-administered rats including rupture of the double membrane, fragmentation of the cristae, and vacuolation (Figure 6B). Meanwhile, the rat group co-administered with donepezil or CA (200, 400, and 800 mg/kg/day) showed attenuation of the aforementioned aberrations in the mitochondria. The other mitochondrial abnormalities identified in the study included ruptured mitochondria (Figure 6C) and elongated mitochondria (Figure 6D), which were also attenuated in rat groups co-administered with donepezil and CA.

#### 3.6.2. Nucleus

A representative electron micrograph of prefrontal cortex of all the rat groups showing nucleus is shown in Figure 7. Normal nucleus with double-layered nuclear membrane, nucleoli, and well-distributed chromatin material could be observed in the control group of rats (Figure 7A). Administration of D-gal and AlCl_3_ produced various morphological alterations viz. distorted nuclear membrane, degenerated chromatin and degenerated nucleoli (Figure 7B). In the rat groups co-administered with donepezil or CA (200, 400, and 800) some of the aforementioned morphological aberrations were prevented (Figure 7C–F).

#### 3.6.3. Synapses

The images of the synapses in the prefrontal cortex are shown in Figure 8A. It can be observed that the prefrontal cortex of the control group rats exhibited abundant synapses, synaptic mitochondria, synaptic vesicles, and synaptic complex (collection of synapses in one area). The rats administered with D-gal and AlCl_3_ exhibited a reduced number of synapses and increased number of abnormal synaptic mitochondria (Figure 8B). Co-administration with donepezil and CA ameliorated some of the above-mentioned morphological aberrations (Figure 8C–F).

## 4. Discussion

Preceding studies have reported that co-administration of D-gal and AlCl_3_ in rats is an easy and inexpensive method of inducing AD-like pathologies and cognitive dysfunction. Some of the observed changes that have been documented include cognitive decline [16,37], oxidative stress, cholinergic dysfunction [12], pathological alteration of astrocytes [38], accumulation of beta amyloid and P-tau in the brain [17], and formation of advanced glycation end products (AGEs) [39], among others. In the current study, D-gal- and AlCl_3_-administered rats displayed impaired cognitive abilities, with marked increases of AChE and MDA levels in both hippocampus and cerebral cortex, while the SOD activity was decreased. Further, ultrastructural aberrations were also observed in the neurons of the prefrontal cortex of the rats’ brains. Hence, the current data proposed that D-gal- and AlCl_3_-induced rat model could serve as a good alternative for the study of AD-related pathologies and for anti-AD drugs screening.

Behavioural responses in rats depends on external stimuli such as past experience and endogenous factors such as gender, age and physiological state. The cyclical changes of sex hormones like oestrogen, progesterone and prolactin in female rats influence their emotional and cognitive functions [40,41,42,43]. These hormonal effects might be confounding factor when testing effects of pharmacological substances on behaviour of female rats. Hence, this study used healthy male albino wistar rats in order to test the effects of CA on their cognitive behaviour after being exposed to D-gal and AlCl_3_. In the present study, rats chronically co-administered with D-gal and AlCl_3_ showed a cognitive decline in MWM test. This is in agreement with preceding works which have reported cognitive impairments in rats administered with D-gal and AlCl_3_ [43,44]. There was no decrease in latency to locate the submerged escape platform by D-gal- and AlCl_3_-induced rats as the training days progressed, whereas co-administration with CA to these rats attenuated the cognitive impairment induced by D-gal and AlCl_3_ in rats as demonstrated by decreased latency to locate the submerged escape platform. Additionally, rats co-administered with CA spent more time in the target quadrant searching for the removed escape platform during the probe trial component of the MWM. It is worthy to note that the cognition-enhancing ability of CA is comparable to that of donepezil, as no differences were observed between the CA groups and donepezil-administered group of rats.

The similarities and differences of primary and tertiary structures of AChEs between species is crucial when selecting AChE inhibitory specificity. However, it is only possible to differentiate between mammals and insects, due to the very high structural, functional, and evolutionary similarities of AChEs among mammals [45]. The tertiary structures of AChEs from *Felis silvestris*, *Bos taurus*, *Oryctolagus cuniculus*, and *Rattus norvegicus* have been predicted to be very similar to that of human structure because the identity of the alignment of their primary sequences with that of humans is very high [45]. Previous studies on anti-AD therapeutics that measured AChE level in the brain used male albino wistar rats [16,46,47]. Therefore, in the present study, male albino wistar rats were preferred to test the effects of CA on AChE levels. ACh is implicated in numerous neuropsychiatric disorders and plays a vital role in cognitive functions [48,49]. The cognitive decline observed in AD has been linked to degeneration of cholinergic neurons in the cerebral cortex and hippocampus, which subsequently resulted in a deficit of cholinergic neurotransmission [12]. In the present study, an impaired cholinergic system was observed in D-gal- and AlCl_3_-administered rats as shown by increased AChE levels in their cerebral cortex and hippocampus. Co-administration of CA with D-gal and AlCl_3_ significantly reduced AChE levels in the rats’ brains. Inhibitory effects of CA on AChE activity was reported by Arora [50] in scopolamine-induced amnesic rats. It has also been reported that perindopril lessened the activities of AChE in D-gal- and AlCl_3_-administered rats [12] and streptozotocin-administered rats [51]. Hence, the present study suggest that the cognition-enhancing ability of CA could be due to its ability to decrease the level of AChE in the rat’s brain. However, some researchers have documented that AChE activities were decreased in AD [52,53]. Thus, more studies are needed to know the exact role of AChE in pathogenesis of AD.

The P-tau protein is now receiving much attention in the field of AD research as a potential target for newer therapeutic agents, and this is due to its involvement in synaptic damage and neuronal dysfunction [5]. The current study showed increased levels of P-tau in the cerebral cortex and hippocampus of D-gal- and AlCl_3_-administered rats_._ This result is in conformism with previous studies which have reported similar findings [16,17]. When the D-gal- and AlCl_3_-administered rats were co-administered with donepezil, the P-tau levels in their cerebral cortex and hippocampus were significantly reduced. However, co-administration of CA to D-gal- and AlCl_3_-administered rats did not reduce the levels of P-tau, thus giving the impression that CA is exerting its cognition-enhancing effects probably through a different pathway, rather than through P-tau.

Altered levels of oxidant have been found in patients with AD, which is attributed to either overproduction of oxidant or deficits in antioxidant. Indications from preclinical and clinical studies suggested that oxidative stress is linked with etiopathology of AD [54,55], resulting in mitochondrial dysfunction [56], increased Aβ-mediated neurotoxicity [57], promotion of synaptic dysfunction, and neuron apoptosis [58]. The common oxidants that participate in redox state includes nitric oxide (NO), hydrogen peroxide (H_2_O_2_), hydroxyl radical (OH), hydroxyl anion (OH^−^), and peroxynitrite (ONOO^−^) [59]. The correlation between the end product of lipid peroxidation, such as thiobarbituric acid-reactive species (TBARS), generally considered as MDA levels, presence of senile plaques, antioxidants enzymes, and accumulation of neurofibrillary tangles in the brains of AD patients, have also been well documented [60]. Because of this strong correlation between oxidative stress and cognitive dysfunction, the agents that are capable of modulating the reactive oxygen species (ROS) are thought to play a significant role in the attenuation of cognitive deficits in AD. The present study has shown that there is an increased level of MDA and decreased activities of SOD in the cerebral cortex and hippocampus in D-gal- and AlCl_3_-administered rats. However, the imbalance of the oxidative stress biomarkers observed were reversed when D-gal- and AlCl_3_-administered rats were co-administered with CA. These results are in agreement with preceding studies that have reported on the antioxidative properties of CA [61,62,63].

The prefrontal cortex is the most evolved region of the brain and subserves the highest order for cognitive abilities. However, it is also the most vulnerable region of the brain to the detrimental effects of stress exposure [64]. In the present study, the ultrastructure of the prefrontal cortex of experimental rats was examined in order to assess the protective effects of CA. A number of ultrastructural morphological abnormalities were observed in rats administered with D-gal and AlCl_3_. Prominent abnormalities that were seen include damaged nucleus, disrupted mitochondria, and alterations of synaptic integrity. Similar morphological changes were also observed in the hippocampus of D-gal- and AlCl_3_-administered rats [26]. Nonetheless, co-administration of CA with D-gal and AlCl_3_ to the rats attenuated some of the identified morphological aberrations. Hence, results from the present study suggest that CA could be exerting its neuroprotective properties through the maintenance of neuronal morphology for the optimal function of the brain.

Previous studies have reported a good safety margin of CA in rats at 1000 mg/kg [65,66] and a lethal dose at 2000 mg/kg [66]. In our previous study CA extract at highest dose 800 mg/kg/day for 10 weeks does not affect locomotor activities and speed of the rats as observed from the results of open field test [26]. CA and its phytochemical constituents such as caffeoylquinic acids and tritepenes have been proved to possess a wide range of biological activities beneficial to human health. These activities includes, neuroprotective, neuritogenic, neuroregenerative, synaptogenic and cognitive enhancing abilities [67,68,69]. Using HPLC analysis four major marker compounds have been reported from the CA extract used in this study including, asiatic acid, madecassic acid, asiaticoside and madecassoside [27]. Therefore, the neuroprotective and the cognitive enhancing effects of CA seen in D-gal and AlCl_3_ induced rats could be due to combined or individual effects of these four major compounds quantified from CA.

## 5. Conclusions

This study investigated the protective effects of CA on D-gal and AlCl_3_ induced rats through behavioural, biochemical and morphological assessments. The results demonstrated that CA irrespective of the dose used, improved cognitive abilities in D-gal and AlCl_3_ induced rats with cognitive deficits. This was achieved because CA restored cholinergic dysfunction by decreasing the level of AChE and attenuated oxidative stress by increasing the activities of SOD and decreasing the levels of MDA. Finally, CA ameliorated the cognitive impairments in rats by preventing ultrastructural morphological aberrations of the neurons in their prefrontal cortex (Figure 9). The authors are continuing to undertake follow up in genetic and protein analysis studies to elucidate other mechanisms underlying the neuroprotective and cognition-enhancing effects of CA in D-gal and AlCl_3_ induced rats with cognitive impairment.

## Figures and Tables

**Figure 1 toxics-07-00019-f001:**
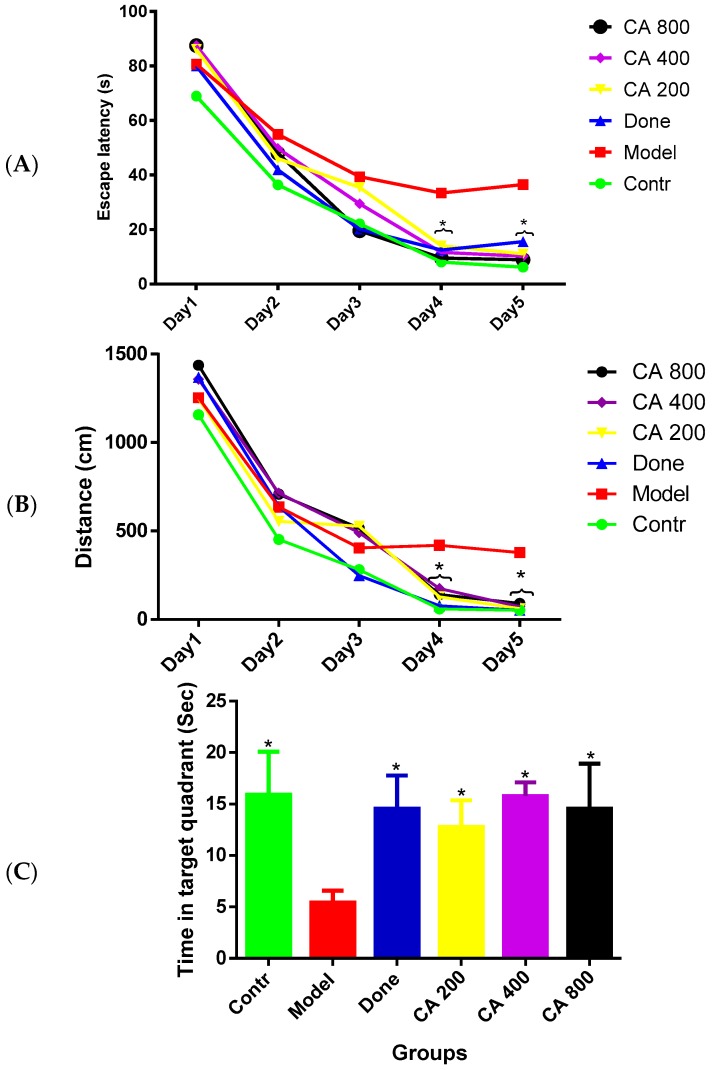
Effects of CA on the behavior of D-gal- and AlCl_3_-induced rats in the Morris water maze (MWM). (**A**) Comparison of escape latency to the submerged escape platform during the five days of training; (**B**) comparison of the distance covered by the rats during the five days of training; and (**C**) comparison of the time spent in the removed escape platform quadrant on day six of the test. Values were presented as mean ± SEM (*n* = 6), * *p* < 0.05 vs. model group.

**Figure 2 toxics-07-00019-f002:**
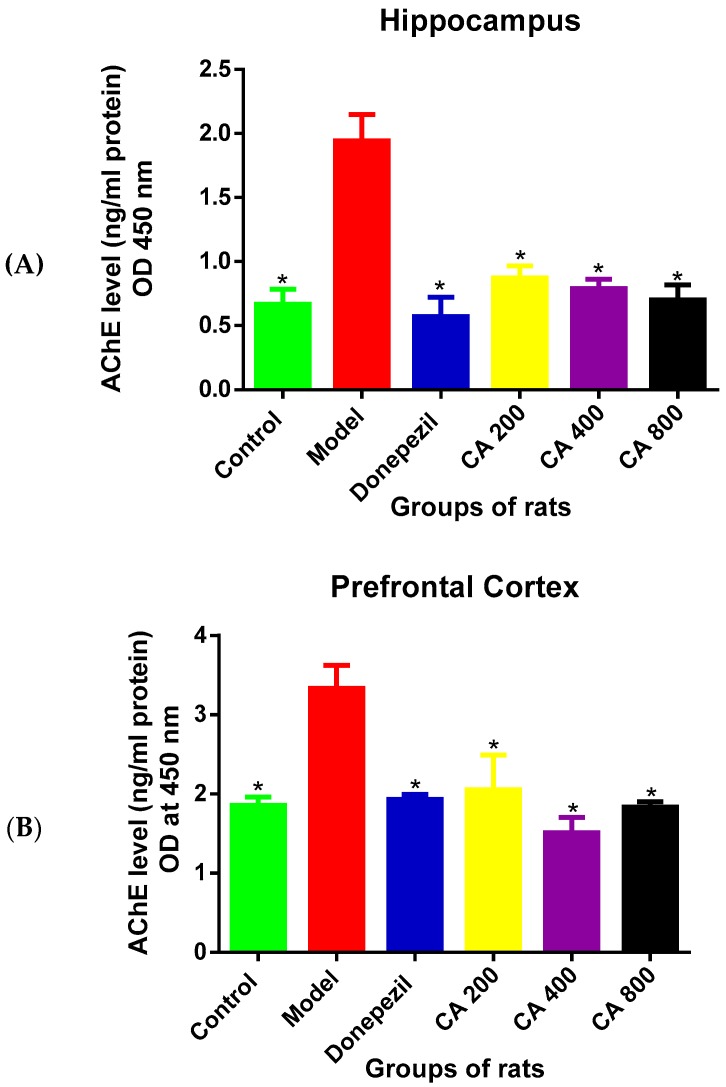
Effects of CA on acetylcholinesterase (AChE) levels. (**A**) Hippocampus, (**B**) cerebral cortex. Data were presented as mean ± SEM, *n* = 3. * *p* < 0.05 vs. model group.

**Figure 3 toxics-07-00019-f003:**
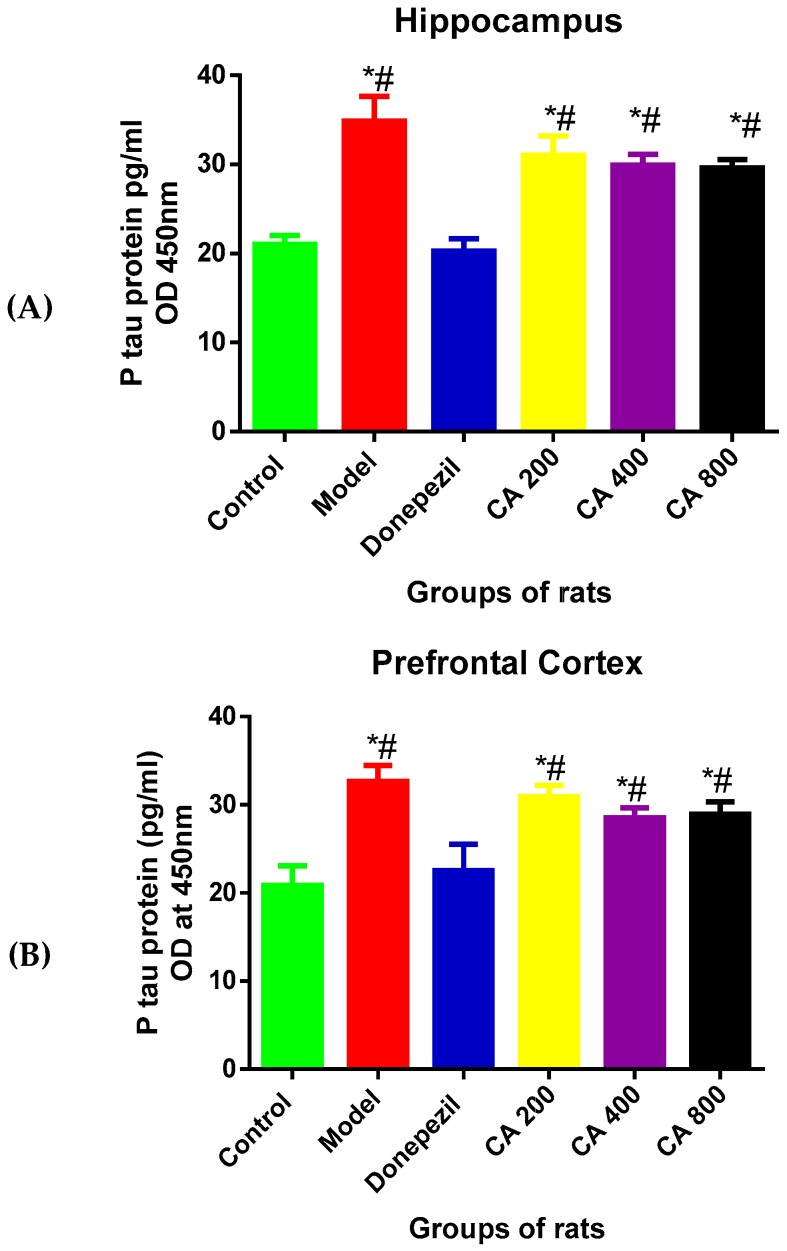
Effects of CA on the levels of P-tau in the hippocampus and cerebral cortex of rat’s administered with D-gal and AlCl_3_. (**A**) Hippocampus, (**B**) cerebral cortex. Data were presented as mean ± SEM, *n* = 3. * *p* < 0.05 vs. control group, # *p* < 0.05 vs. model group.

**Figure 4 toxics-07-00019-f004:**
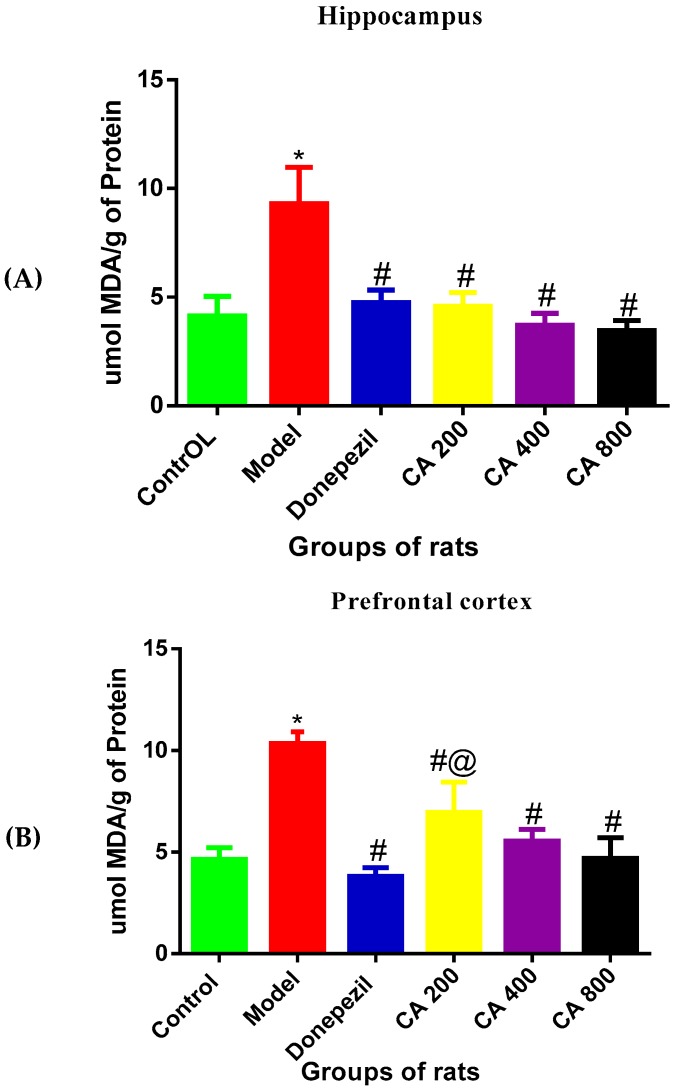
Effects of CA on malondialdehyde (MDA) levels in the hippocampus and cerebral cortex of D-gal- and AlCl_3_-administered rats. (**A**) Hippocampus, (**B**) cerebral cortex. Data were presented as mean ± SEM, *n* = 3. * *p* < 0.05 vs. control group, # *p* < 0.05 vs. model group and @ *p* < 0.05 vs. donepezil.

**Figure 5 toxics-07-00019-f005:**
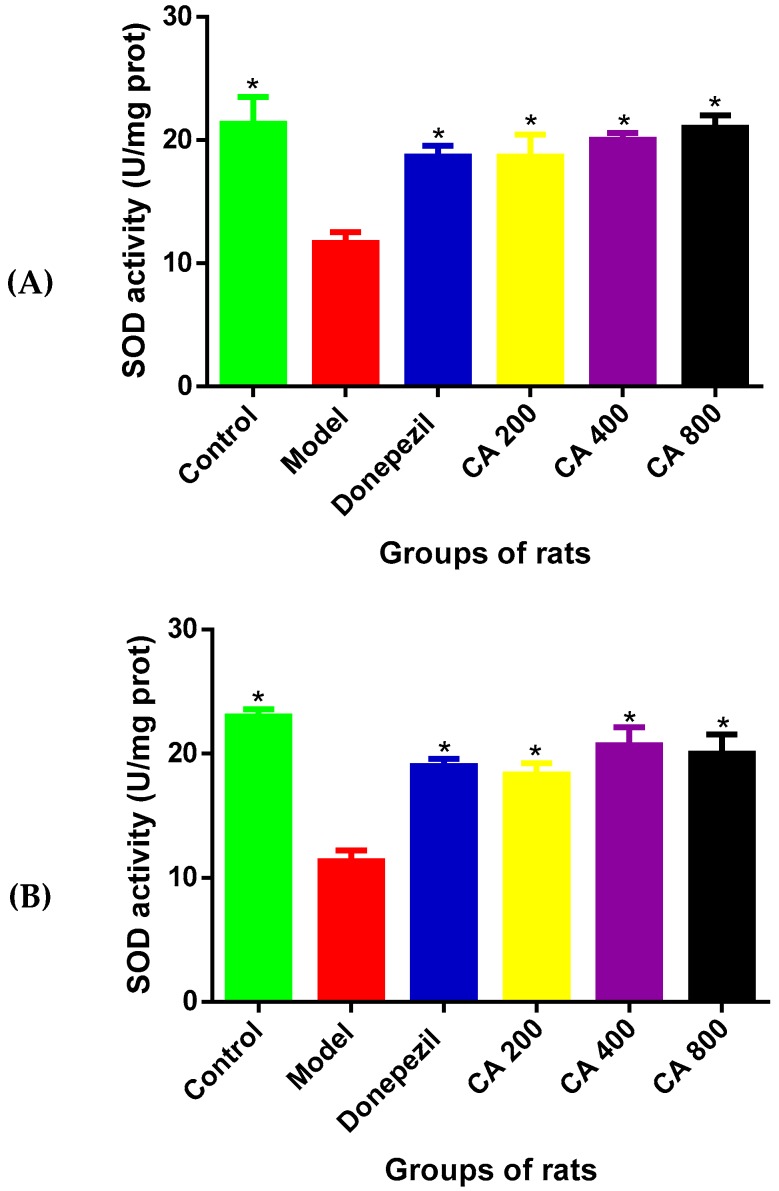
Effects of CA on superoxide dismutase (SOD) activities in the hippocampus and cerebral cortex of D-gal- and AlCl_3_-induced rats. (**A**) Hippocampus, (**B**) cerebral cortex. Values are presented as mean ± SEM, *n* = 3. * *p* < 0.05 vs. model group, # *p* < 0.05 vs. control group.

**Figure 6 toxics-07-00019-f006:**
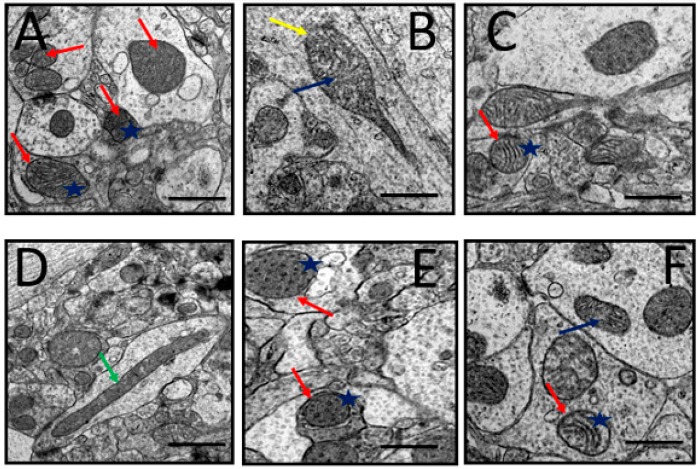
Representative TEM images depicting rat’s prefrontal cortex showing neuronal mitochondrial abnormalities after exposure to D-gal and AlCl_3_ and those co-administered with donepezil and CA. (**A**) Control group: Normal mitochondrion (red arrow) with homogenous dense matrix, organized cristae (blue star), and double mitochondrial membrane. (**B**) Model group: Showed degenerated cristae (blue arrow) and ruptured mitochondrial membrane (yellow arrow). (**C**) Donepezil group: Showed normal and abnormal mitochondria. (**D**) CA 200 group: Showed elongated mitochondria (green arrow). (**E**,**F**) CA 400 and CA 800 groups, respectively: Showed normal mitochondria (red arrows) and well-organized cristae (blue star). Scale bar 0.5 µm.

**Figure 7 toxics-07-00019-f007:**
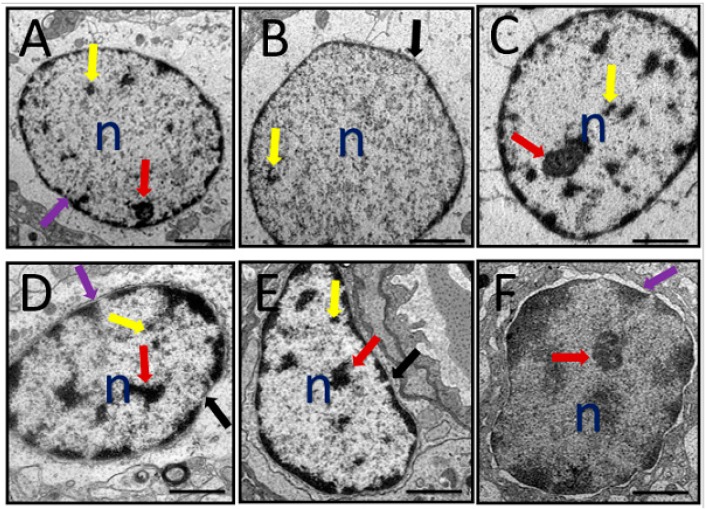
A representative TEM micrograph depicting rat’s prefrontal cortex with identified nucleus abnormalities after administration of D-gal and AlCl_3_ and those co-administered with donepezil and CA. (**A**) Control group: Showing nucleoli (red arrow), chromatin material (yellow arrow), and double layered nuclear membrane (purple arrow). (**B**) Model group: Showing degenerated chromatin material, absence of nucleoli, and a break of nuclear membrane (black arrow). (**C**) Donepezil group: Showing close to normal nucleus. (**D**) CA 200 group: Showing both double-layered nuclear membrane (purple arrow) and a broken nuclear membrane (black arrow), nucleoli (red arrow), and chromatic (yellow arrow). (**E**) CA 400 group: Showing close to normal nucleus but with crescent formation. (**F**) CA 800 group: Showing a normal nucleus. *n* = Nucleus. Scale bar 0.2 µm.

**Figure 8 toxics-07-00019-f008:**
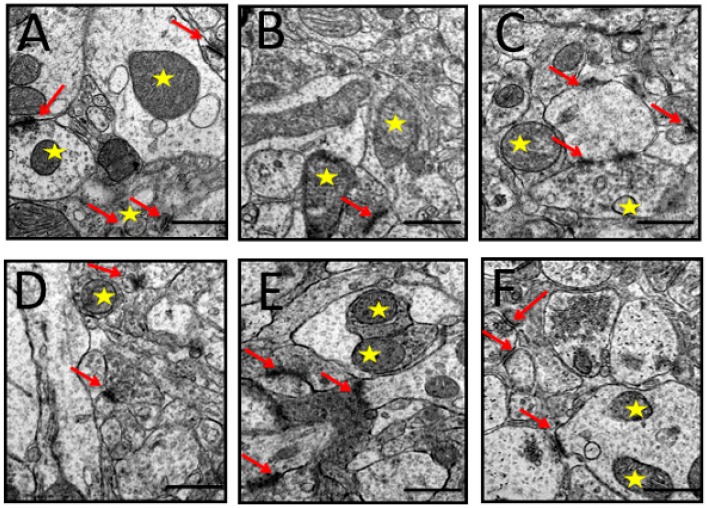
TEM micrographs of synapses identified in the prefrontal cortex of rats induced with D-gal and AlCl_3_ and those co-administered with donepezil and CA. (**A**) Control group: Showed synaptic complex (red arrows) and healthy presynaptic mitochondria (yellow star). (**B**) Model group: Showed a blurry synapse and distorted presynaptic mitochondria. (**C**) Donepezil group: Showed synaptic complex. (**D**–**F**) Showed a good number of synapses (red arrows), synaptic vesicles, and presynaptic mitochondria (yellow star). (Magnification 40,000×).

**Figure 9 toxics-07-00019-f009:**
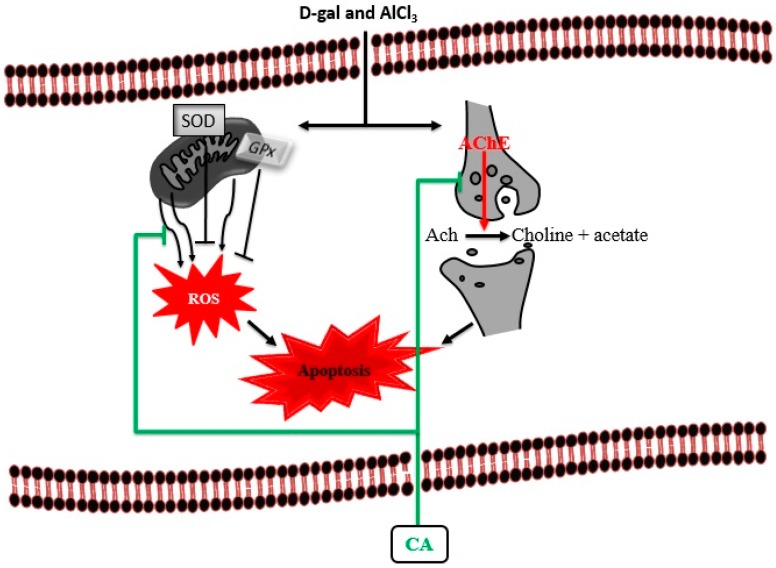
Proposed cognition-enhancing mechanism of CA in D-gal and AlCl_3_ induced rats. Administration of D-gal/AlCl_3_ induced oxidative stress, cholinergic dysfunction and apoptosis. Administration of CA decreased the levels of AChE and MDA by blocking their activities. CA also increased the levels of SOD and maintain the neuronal cytoarchitecture by prevention of apoptosis.

**Table 1 toxics-07-00019-t001:** AlCl3-, D-gal-, and *Centella asiatica* (CA)-treated groups and the control.

Groups	Description	Treatment i.p	Treatment p.o
I	Control	Saline	Distilled water
II	Model	D-gal 60 mg/kg.bwt	AlCl_3_ 200 mg/kg.bwt
III	Donepezil	D-gal 60 mg/kg.bwt	AlCl_3_ 200 mg/kg.bw + Done 1 mg/kg.bwt
IV	CA 200	D-gal 60 mg/kg.bwt	AlCl_3_ 200 mg/kg.bw + CA 200 mg/kg.bwt
V	CA 400	D-gal 60 mg/kg.bwt	AlCl_3_ 200 mg/kg.bw + CA 400 mg/kg.bwt
VI	CA 800	D-gal 60 mg/kg.bwt	AlCl_3_ 200 mg/kg.bw + CA 800 mg/kg.bwt

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
