# Peer review of "Protective Effects of Centella asiatica on Cognitive Deficits Induced by D-gal/AlCl3 via Inhibition of Oxidative Stress and Attenuation of Acetylcholinesterase Level"

_toxics, 2019, doi:10.3390/toxics7020019_

Round 1
Reviewer 1 Report
Dear Editor,
I am sending you my referee report on the manuscript entitled: Centella asiatica protects against cognitive deficits induced by D‐galactose and aluminium chloride via inhibition of lipid peroxidation and acetylcholinesterase activity (by: Samaila Musa Chiroma et al.) originally submitted to TOXICS.
Bellow are my few remarks and suggestions:
-the choice of male albino Wistar rats should be discussed (why rats? why not female?) - mostly in in vivo test there are known species differences, especially when cholinesterase’s are discussed (Wiesner et al. JEnzymeInhibMedChem 2007, 22(4), 417). The choice should be discussed
-what was the purity of donepezil, how was the purity checked prior the experiment?
Author Response
Reviewer 1
Point 1
-the choice of male albino Wistar rats should be discussed (why rats? why not female?) - mostly in in vivo test there are known species differences, especially when cholinesterase’s are discussed (Wiesner et al. JEnzymeInhibMedChem 2007, 22(4), 417). The choice should be discussed
Author’s response 1
The choice of male albino wistar rats over females and the species differences in relations to cholinesterases were all discussed as recommended, thank you for the wonderful article you recommended.
“Behavioural responses in rats depends on external stimuli such as past experience and endogenous factors such as gender, age and physiological state. The cyclical changes of sex hormones like oestrogen, progesterone and prolactin in female rats influence their emotional and cognitive functions (Zapata et al., 2015) (Lovick, 2006; Mora et al., 1996; Paris and Frye, 2008). These hormonal effects might be confounding factor when testing effects of pharmacological substances on behaviour of female rats. Hence, this study used healthy male albino wistar rats in order to test the effects of CA on their cognitive behaviour after being exposed to D-gal and AlCl3.”
“The similarities and differences of primary and tertiary structures of AChEs between species is crucial when selecting AChE inhibitory specificity. However, it is only possible to differentiate between mammals and insects, due to the very high structural, functional and evolutionary similarities of AChE’s among mammals (Wiesner et al., 2007). The tertiary structures of AChEs from Felis silvestris, Bos taurus, Oryctolagus cuniculus and Rattus norvegicus have been predicted to be very similar to that of human structure because the identity of the alignment of their primary sequences with that of human is very high (Wiesner et al., 2007). Previous studies on anti-AD therapeutics that measured AChE activities used male abino wistar rats (Hasanein and Mahtaj, 2015; Mathiyazahan et al., 2015; Zhang et al., 2016). Therefore, the present study also used male abino wistar rats to test the effects on CA on AChE levels.”
Point 2
-what was the purity of donepezil, how was the purity checked prior the experiment?
Author’s response 1
We got donepezil from a reputable company Esai Co. Ltd. (Tokyo, Japan) who has a certificate of Good Manufacturing Practice, the purity of donepezil from the leaflet is 91.2%. With that, we do not go further to check the purity of the drug but rather, we kept it at the recommended temperature and away from lights then used it for our research.
References
Hasanein, P., Mahtaj, A.K., 2015. Ameliorative effect of rosmarinic acid on scopolamine-induced memory impairment in rats. Neurosci. Lett. 585, 23–27. doi:10.1016/j.neulet.2014.11.027
Lovick, T.A., 2006. Plasticity of GABAA receptor subunit expression during the oestrous cycle of the rat: implications for premenstrual syndrome in women. Exp. Physiol. 91, 655–660.
Mathiyazahan, D.B., Justin Thenmozhi, A., Manivasagam, T., 2015. Protective effect of black tea extract against aluminium chloride-induced Alzheimer’s disease in rats: A behavioural, biochemical and molecular approach. J. Funct. Foods 16, 423–435. doi:10.1016/j.jff.2015.05.001
Mora, S., Dussaubat, N., Díaz-Véliz, G., 1996. Effects of the estrous cycle and ovarian hormones on behavioral indices of anxiety in female rats. Psychoneuroendocrinology 21, 609–620.
Paris, J.J., Frye, C.A., 2008. Estrous cycle, pregnancy, and parity enhance performance of rats in object recognition or object placement tasks. Reproduction 136, 105–115.
Wiesner, J., Kříž, Z., Kuča, K., Jun, D., Koča, J., 2007. Acetylcholinesterases–the structural similarities and differences. J. Enzyme Inhib. Med. Chem. 22, 417–424.
Zapata, M.P., Rodríguez Echandía, E.L., González Jatuff, A.S., Quercetti, M., Torrecilla, M., 2015. Influence of the estrous cycle on some non reproductive behaviors and on brain mechanisms in the female rat. Interdiscip. Rev. Psicol. y Ciencias Afines 29, 63–77. doi:10.16888/interd.2012.29.1.4
Zhang, Y., Yang, X., Jin, G., Yang, X., Zhang, Y., 2016. Polysaccharides from Pleurotus ostreatus alleviate cognitive impairment in a rat model of Alzheimer’s disease. Int. J. Biol. Macromol. 92, 935–941. doi:10.1016/j.ijbiomac.2016.08.008
Reviewer 2 Report
Overall, this paper presents a study examining CA extract against cognitive deficits in rats induced by D-gal and AlCl3. The authors use D-gal and AlCl3 to induce cognitive deficits similar to Alzheimer's disease.They then evaluate whether the CA extract at varying doses can alleviate these deficits. Cognitive function is measured through water maze test. AChE, P-tau, and MDA levels in the brain are also used to assess the treatment, as well as TEM imaging of mitochondria and nuclei in brain cells. Donepezil is used at a control. The authors conclude that the CA extract improved cognitive function, decreased AChE activity, reduced MDA, and prevented morphological alterations.
Broadly, the strength of this paper lies in providing a potential new lead for Alzheimer's drug development. However, some weaknesses have been identified that need to be addressed.
1) There were numerous spelling and grammatical errors throughout the paper that should be addressed by additional proofreading. For example, in the abstract alone:
- Line 25, neurodegenerative is misspelled
- Line 30, "were" should be "was"
- Line 32, "were" should be "was"
- Line 36, "a" should be omitted
2) How were the dosages chosen? For example:
- Donepezil was dosed at 1mg/kg/day. This would equate to 70mg/day for a 70kg human. However, clinically it is dosed at ~10 mg/day.
- The lowest dose of CA extract was 200 mg/kg/day. This would equate to 14g/day for a 70 kg human. This is a huge dose. How is this useful therapeutically?
3) Why was this model for inducing cognitive deficits chosen? What about using a transgenic rat model that mimics Alzheimer's disease?
4) The explanation for looking at P-tau and MDA levels is given in the results. This should be mentioned in the introduction to clarify for readers.
5) The claim that CA extract decreased AChE activity is not fully supported by the data. The ELISA assay used measured amount of AChE present. Activity was not measured directly, and the assumption is being made that decreased enzyme levels are giving decreased activity. The authors can say that AChe levels are decreased. However, AChE activity would need to be assessed by another assay (e.g. Ellman assay).
6) Has any effort been made to identify the active compound in the CA extract? This would make this study much more appealing to medicinal chemists as the active compound could then be derivatized to assess structure activity relationship. At the very least, some speculation could be made as to potential active components of the extract.
Author Response
Reviewer 2
Broadly, the strength of this paper lies in providing a potential new lead for Alzheimer's drug development. However, some weaknesses have been identified that need to be addressed.
Point 1
1) There were numerous spelling and grammatical errors throughout the paper that should be addressed by additional proofreading. For example, in the abstract alone:
- Line 25, neurodegenerative is misspelled
- Line 30, "were" should be "was"
- Line 32, "were" should be "was"
- Line 36, "a" should be omitted
Author’s response 1
The whole article was proofread.
Point 2
2) How were the dosages chosen? For example:
Point 2a
- Donepezil was dosed at 1mg/kg/day. This would equate to 70mg/day for a 70kg human. However, clinically it is dosed at ~10 mg/day.
Author’s response 2a
The dose of donepezil (1mg/kg/day) was adopted from (Song et al., 2017). A dose of 5mg/kg of donepezil was reported by (Biswas et al., 2018) and there was no negative effects, 0.75mg/kg of donepezil by (Tripathi et al., 2009) and 2mg/kg by (Hasanein and Mahtaj, 2015) were also used and effective. We adopted 1mg/kg because it’s a recent study.
Point 2b
- The lowest dose of CA extract was 200 mg/kg/day. This would equate to 14g/day for a 70 kg human. This is a huge dose. How is this useful therapeutically?
Author’s response 2b
We are used crude extract of CA (Reference number: AuRins-MIA-1-0) (Binti Mohd Yusuf Yeo et al., 2018; Wong et al., 2019) supplied by our collaborators. The dose chosen for this study was based on unpublished data from our collaborators and some literatures researched. Previous studies have reported a good safety margin of CA in rats at 1000mg/kg (Chivapat et al., 2011; Deshpande et al., 2015) and a lethal dose at 2000mg/kg (Deshpande et al., 2015). 70 days administration of CA at 200mg/kg have ameliorated lead induced toxicity in rats (Sainath et al., 2011), neuroprotective and cognitive enhancing properties of CA at different at doses ranging from 15, 100, 300, 500 and 600mg/kg in differenent pathological context was also reported by several authors (Gray et al., 2016; Nampoothiri et al., 2015; Oruganti et al., 2010; Syed Umesalma, 2015; Wong et al., 2019). Our study used CA at doses of 200, 400 and 800mg/kg for the first time in D-gal/AlCl3 induced rats.
Point 3
3) Why was this model for inducing cognitive deficits chosen? What about using a transgenic rat model that mimics Alzheimer's disease?
Author’s response 3
Based on the expression of human genes relevant to early-onset AD, a wide range of transgenic rats were created since early 2000s, including wild type mutated PS1 or mutated APP (Carmo and Cuello, 2013). Due to their nature transgenic animal models only relate to the early onset AD which represents only 5% of AD diagnosis in humans while the remaining 95% are late onset AD which the cause is yet to be determined. Even though both late onset AD and (Lecanu and Papadopoulos, 2013a) early onset AD have similar clinical presentations, late onset AD does not involve any form of gene mutation and the reason for the amyloid accumulation and aggregation still remain elusive. Consequently, transgenic rat models are deemed to be unsuited for unveiling the etiopathology of late onset AD (Lecanu and Papadopoulos, 2013b).
The biggest advantage of choosing chemically induced models such as D-gal/AlCl3 is that, it mimics the pathogenesis of AD and produce the late onset AD which is 95% form of AD diagnosis (Malekzadeh et al., 2017).
Further, transgenic rat model are expensive and not readily available at our locality. Finally CA has been reported to have improved behavioural deficits in the Tg2576 mouse, a trangenic model of AD with high beta amyloid burden (Soumyanath et al., 2012), hence the present study used CA on non-transgenic model.
Point 4
4) The explanation for looking at P-tau and MDA levels is given in the results. This should be mentioned in the introduction to clarify for readers.
Author’s response 4
The explanations were removed from results and moved to introduction as suggested.
Point 5
5) The claim that CA extract decreased AChE activity is not fully supported by the data. The ELISA assay used measured amount of AChE present. Activity was not measured directly, and the assumption is being made that decreased enzyme levels are giving decreased activity. The authors can say that AChe levels are decreased. However, AChE activity would need to be assessed by another assay (e.g. Ellman assay).
Author’s response 5
AChE activities has been changed to AChE level as rightly explained by the reviewer. Three days given to us by the editorial board to answer the reviewers’ comments is too short to carryout Ellman assay for AChE activites as we do not have the reagents at present.
Point 6
6) Has any effort been made to identify the active compound in the CA extract? This would make this study much more appealing to medicinal chemists as the active compound could then be derivatized to assess structure activity relationship. At the very least, some speculation could be made as to potential active components of the extract.
Author’s response 6
Our collaborators from other institution have quantified four marker compounds from CA table 1 (Wong et al., 2019).
Table 1. Quantification of marker compounds in CA extract (mg/g) using HPLC analysis.
Compound | RT (min) | Content (mg/g) | SD | Content ± SD (mg/g) | |
Madecassoside Asiaticoside Madecassic acid Asiatic acid | 15.435 16.518 24.865 28.408 | 0.1161 0.1411 0.1437 0.0725 | 0.0003 0.0019 0.0008 0.0002 | 0.1161 ± 0.0003 0.1411 ± 0.0019 0.1437 ± 0.0008 0.0729 ± 0.0002 |
References
Binti Mohd Yusuf Yeo, N.A., Muthuraju, S., Wong, J.H., Mohammed, F.R., Senik, M.H., Zhang, J., Yusof, S.R., Jaafar, H., Adenan, M. llham, Mohamad, H., Tengku Muhammad, T.S., Abdullah, J.M., 2018. Hippocampal amino-3-hydroxy-5-methyl-4-isoxazolepropionic acid GluA1 (AMPA GluA1) receptor subunit involves in learning and memory improvement following treatment with Centella asiatica extract in adolescent rats. Brain Behav. 8, 1–14. doi:10.1002/brb3.1093
Biswas, J., Gupta, S., Verma, D.K., Gupta, P., Singh, A., Tiwari, S., Goswami, P., Sharma, S., Singh, S., 2018. Involvement of glucose related energy crisis and endoplasmic reticulum stress: Insinuation of streptozotocin induced Alzheimer’s like pathology. Cell. Signal. 42, 211–226. doi:10.1016/j.cellsig.2017.10.018
Carmo, S. Do, Cuello, A.C., 2013. Modeling Alzheimer ’ s disease in transgenic rats 1, 1–11.
Chivapat, S., Chavalittumrong, P., Tantisira, M.H., 2011. Acute and sub-chronic toxicity studies of a standardized extract of Centella asiatica ECa 233. Thai J. Pharm. Sci. 35, 55–64.
Deshpande, P.O., Mohan, V., Thakurdesai, P., 2015. Preclinical Safety Assessment of Standardized Extract of Centella asiatica (L.) Urban Leaves. Toxicol. Int. 22, 10–20. doi:10.4103/0971-6580.172251
Gray, N.E., Harris, C.J., Quinn, J.F., Soumyanath, A., 2016. Centella asiatica modulates antioxidant and mitochondrial pathways and improves cognitive function in mice. J. Ethnopharmacol. 180, 78–86. doi:10.1016/j.jep.2016.01.013
Hasanein, P., Mahtaj, A.K., 2015. Ameliorative effect of rosmarinic acid on scopolamine-induced memory impairment in rats. Neurosci. Lett. 585, 23–27. doi:10.1016/j.neulet.2014.11.027
Lecanu, L., Papadopoulos, V., 2013a. Modelin Alzheimer’s disease with non-transgenic rat models 1–9. doi:10.1074/jbc.M109.083915
Lecanu, L., Papadopoulos, V., 2013b. Modeling Alzheimer ’ s disease with non-transgenic rat models. Alzheimer’s Dement. 5, 1–10.
Malekzadeh, S., Edalatmanesh, M.A., Mehrabani, D., Shariati, M., Science, F., Branch, S., Cell, S., Technology, T., Branch, K., 2017. Drugs Induced Alzheimer’s Disease in Animal Model 6, 185–196. doi:10.22086/gmj.v6i3.820
Nampoothiri, M., John, J., Kumar, N., Mudgal, J., Nampurath, G.K., Chamallamudi, M.R., 2015. Modulatory role of simvastatin against aluminium chloride-induced behavioural and biochemical changes in rats. Behav. Neurol. 2015. doi:10.1155/2015/210169
Oruganti, M., Roy, B.K., Singh, K.K., Prasad, R., Kumar, S., 2010. Safety Assemment of Centella asiatica in Albino Rats. Pharmacogn. J. 2, 5–13.
Sainath, S.B., Meena, R., Supriya, C., Reddy, K.P., Reddy, P.S., 2011. Protective role of Centella asiatica on lead-induced oxidative stress and suppressed reproductive health in male rats. Environ. Toxicol. Pharmacol. 32, 146–154. doi:10.1016/j.etap.2011.04.005
Song, X., Liu, B., Cui, L., Zhou, B., Liu, W., Xu, F., Hayashi, T., Hattori, S., Ushiki-Kaku, Y., Tashiro, S. ichi, Ikejima, T., 2017. Silibinin ameliorates anxiety/depression-like behaviors in amyloid β-treated rats by upregulating BDNF/TrkB pathway and attenuating autophagy in hippocampus. Physiol. Behav. 179, 487–493. doi:10.1016/j.physbeh.2017.07.023
Soumyanath, A., Zhong, Y.-P., Henson, E., Wadsworth, T., Bishop, J., Gold, B.G., Quinn, J.F., 2012. Centella asiatica extract improves behavioral deficits in a mouse model of Alzheimer’s disease: investigation of a possible mechanism of action. Int. J. Alzheimer’s Dis. 2012.
Syed Umesalma, S.A., 2015. Protective Effect of Centella asiatica against Aluminium-Induced Neurotoxicity in Cerebral Cortex, Striatum, Hypothalamus and Hippocampus of Rat Brain- Histopathological, and Biochemical Approach. J. Mol. Biomark. Diagn. 06. doi:10.4172/2155-9929.1000212
Tripathi, S., Mahdi, A.A., Nawab, A., Chander, R., Hasan, M., Siddiqui, M.S., Mahdi, F., Mitra, K., Bajpai, V.K., 2009. Influence of age on aluminum induced lipid peroxidation and neurolipofuscin in frontal cortex of rat brain: A behavioral, biochemical and ultrastructural study. Brain Res. 1253, 107–116. doi:10.1016/j.brainres.2008.11.060
Wong, J.H., Muthuraju, S., Reza, F., Senik, M.H., Zhang, J., Mohd Yusuf Yeo, N.A.B., Chuang, H.G., Jaafar, H., Yusof, S.R., Mohamad, H., Tengku Muhammad, T.S., Ismail, N.H., Husin, S.S., Abdullah, J.M., 2019. Differential expression of entorhinal cortex and hippocampal subfields α-amino-3-hydroxy-5-methyl-4-isoxazolepropionic acid (AMPA) and N-methyl-D-aspartate (NMDA) receptors enhanced learning and memory of rats following administration of Centella asiatica. Biomed. Pharmacother. 110, 168–180. doi:10.1016/j.biopha.2018.11.044
